# Smart Electric Vehicle Charging via Adjustable Real-Time Charging Rates

Theron Smith [1,*] , Joseph Garcia [1] and Gregory Washington [2]

1  Department of Mechanical and Aerospace Engineering, University of California, Irvine, CA 92697-3550, USA; josephg2@uci.edu
2  George Mason University, Fairfax, VA 22030, USA; gwashin@gmu.edu
*  Correspondence: smithtf@uci.edu; Tel.: +1-510-703-4091

**Abstract:** This paper presents a plug-in electric vehicle (PEV) charging control algorithm, Adjustable Real-Time Valley Filling (ARVF), to improve PEV charging and minimize adverse effects from uncontrolled PEV charging on the grid. ARVF operates in real time, adjusts to sudden deviations between forecasted and actual baseloads, and uses fuzzy logic to deliver variable charging rates between 1.9 and 7.2 kW. Fuzzy logic is selected for this application because it can optimize nonlinear systems, operate in real time, scale efficiently, and be computationally fast, making ARVF a robust algorithm for real-world applications. In addition, this study proves that when the forecasted and actual baseload vary by more than 20%, its real-time capability is more advantageous than algorithms that use optimization techniques on predicted baseload data.

**Keywords:** fuzzy logic; plug-in electric vehicle; valley filling; distribution transformer; electric vehicle charging; smart charging





## 1. Introduction

As cities expand and populations increase, the amount of available fossil fuels decreases and air quality laws are made stricter [1]. Thus, Zero-Emission Vehicles (ZEV) become a more appealing answer to future transportation problems [2–4]. California has initiated measures to encourage the adoption of ZEVs, to have at least 5 million ZEVs on the road by 2030 to reduce emissions from transportation sources [5]. Battery electric vehicles (BEVs), fuel cell electric vehicles (FCEVs), and plug-in electric vehicles (PEVs) are the three types of ZEVs [6]. Since 2011, almost two million PEVs have been sold, with California accounting for nearly one-third of all sales, shown in Figure 1 [7–9]. However, increased PEV adoption will lead to disturbances in the electric grid because of power demands exceeding the grid's initial design conditions [10–14].

The increased electric demand from PEV charging can easily double or even triple a household's energy demand [15,16]. Furthermore, the load from multiple charging PEVs can shorten transformer life and necessitate repairs and replacements of parts earlier than expected [15,17]. These concerns led to the development of "smart charging" protocols [18]. These strategies allow PEV adoption to expand and manage electric grid levels within safe limits, reducing the need for utility investments to renovate the distribution network to accommodate larger demand loads [19–22]. Without smart charging protocols, vehicles immediately begin charging when plugged in, regardless of the time or demand, which is referred to as "uncontrolled charging" [23]. By shifting charging to the late evenings and early mornings, a practice known as "valley filling" is used to control PEV charging [19,24,25]. This technique charges vehicles when the grid's energy demand is low, lessening the impact of the extra load caused by charging PEVs. Multiple charging control systems have been created to regulate PEV charging, with decentralized and centralized control being the most common [26,27].

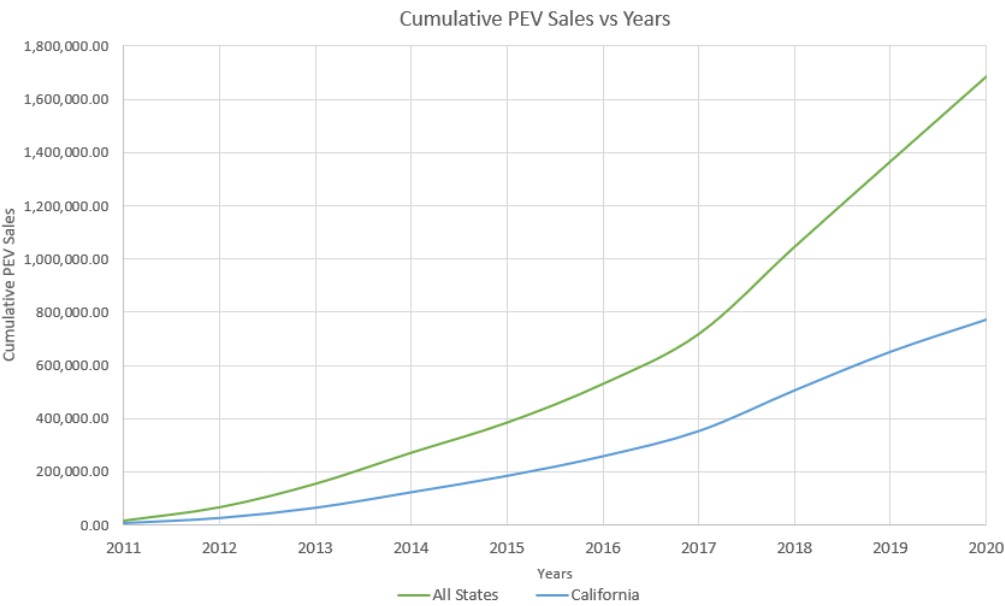

**Figure 1.** Cumulative PEV sales in the U.S. from 2011 to 2020.

Centralized smart charging methods rely on a central operating controller to manage charging patterns for customers [28–32]. The controller lowers the demand load by estimating the ideal charging profile for each PEV based on the customer's requested charge, residence duration, and vehicle plug-in time, as well as the predicted load [1,28,33–36]. Several centralized charging methods that incorporate fuzzy logic have been produced. Masoum et al. [35] developed a centralized method employing maximum sensitivity selection (MSS) optimization and fuzzy logic to optimize the algorithm. Hajforoosh et al. [37] improved on Masoum et al. [35] and provided two solutions for reducing energy generation costs and grid losses while increasing electricity delivered to automobiles. Fuzzy logic is used by Hajforoosh et al. [37] to improve the effectiveness of the genetic algorithm and discrete particle swarm optimization used in the study. However, this approach is not very practical because vehicles cannot input a plug-out time. In another study, Singh et al. [38] implement fuzzy logic controllers in PEV charging stations at distribution substations. The controller that is applied to PEV charging stations regulates the amount of power each station receives. The controller installed in the substation monitors the total amount of power consumed by all the charging stations connected to it. This protocol demonstrates that it can peak-shave, load flatten, and valley-fill charging profiles; however, this strategy relies on vehicle-to-grid (V2G) technology. Unfortunately, Bishop et al. [39] show that V2G technology accelerates wear and the frequency that PEVs need battery replacements.

Several developed decentralized valley-filling algorithms [15,19,28,40] have gained interest due to their ability to allow PEVs to establish individual charging patterns. Users can create their own charging patterns, which can be based on a priced or non-priced electric scheme [40,41]. These profiles are sent to a centralized operating controller that updates the demand load, adding the newly constructed PEV charging pattern [42,43]. Unfortunately, there is not a guarantee that the overall structure of the demand load will be optimal when using this approach [28,44,45]. Zhang et al. [19] introduced a valley-filling algorithm that generates a near-ideal solution, but it produces significant harmful transformer overload and overheating. Ramos Muñoz et al. [46] expand this strategy by using Modified Timeslot Rejection (MTR) to reduce stress on transformers.

The MTR protocol uses forecasted data to determine the optimal times to charge vehicles and establishes vehicle charging patterns based on this information. When a vehicle plugs into the network, the algorithm assesses the forecasted baseload and determines the optimal time for the vehicle to charge using the grid valley-filling algorithm by Zhang et al. [19]. After the profile is created, the algorithm verifies that the profile does

not violate any of the recommended temperature and loading limits in the IEEE C57.91 standard [47]. If the profile does exceed any transformer operating limits, the profile is adjusted to minimize stress on the transformer. The algorithm then updates the charging profile for the vehicle and sends it to the grid operator to update the projected demand load. The process is then repeated for the following vehicle that plugs in.

MTR performs well when the forecasted and actual baseloads are extremely similar; however, its functionality decreases significantly when the predicted and actual baseloads deviate. In reality, demand varies from day to day. While historical data may provide a relative sense of what to expect, it can never be exact, especially given the dramatic rise in PEVs and BEVs nationally. Algorithms that are more robust to these changes without requiring costly additional optimization are needed. As technology evolves, the need for an algorithm that can adjust for variation will intensify as the number and power demand of devices connecting to the grid will continue to increase [48]. These changes will lead to more frequent and larger fluctuations in baseloads, making an algorithm that can adjust for variation valuable.

Additionally, MTR only optimizes transformer loads that violate the operating limits. If all transformer loads were assessed, not only when operating boundaries are violated, transformer life can be extended further as loss of transformer life is directly related to the load transformers observe. Focusing on local distribution transformers is important because a single PEV home-charger can draw as much as 11.5 kW, enabling a few simultaneously charging PEVs during an early summer evening to dramatically degrade transformer lifespan [33,49]. Razeghi et al. show that extreme uncontrolled charging conditions can increase the loss of life percentage for transformers by over 2000% in as little as 24 h [15]. Furthermore, residential transformers should be monitored more heavily, as they are the component most susceptible to damage from uncontrolled damage within the U.S. power system [50].

This work presents Adjustable Real-Time Valley Filling (ARVF), a practical valley-filling strategy that determines its solutions in real time. ARVF extends the analysis of MTR by Ramos Muñoz et al. [46], focusing on improving PEV charging at the local power distribution level and minimizing excess damage to distribution transformers, a rationale from Q. Gong et al. [50]. The objectives of this study are (1) to create a robust strategy that can adjust to significant unforeseen variations in the forecasted baseload and (2) to investigate and determine the amount of variation in the baseload that causes algorithms such as Zhang et al. [19] and Ramos Muñoz et al. [46] to no longer produce optimal solutions using forecasted data.

To react to sudden changes in the baseload, ARVF needs to be capable of changing the charging rates it is delivering to vehicles. For example, suppose the baseload suddenly increases. In that case, the algorithm can decrease the rate that is being delivered to a vehicle. If the baseload suddenly decreases, the algorithm can increase the rate that is being delivered to a vehicle. This attribute is implemented in this study by assuming ARVF can charge vehicles at any rate between 1.9 and 7.2 kW. Fuzzy logic is used to determine the rate delivered to vehicles because of its inheritability to tolerate the concept of partial truth, where the truth value may range between completely true and completely false, much like the range of the rates administered to charging vehicles [50,51]. This allows ARVF to deliver rates that range from 1.9 to 7.2 kW in a continuous spectrum to vehicles based on their needs. In addition, fuzzy logic is selected for this application because it can optimize nonlinear systems, operates in real time, scales efficiently, and is not computationally expensive. These characteristics are important for an algorithm being used in real-world applications.

## 2. Materials and Methods

### 2.1. Problem Formulation

In this study, a controller is assumed to be attached to distribution transformers and strategically vary the charging rate supplied to each vehicle. The controller enables an

aggregator to use smart metering technology to access PEV information, such as when a PEV is plugging into the grid, requesting charging and dwelling duration.

The algorithm uses fuzzy logic to produce a valley-filling effect by monitoring when distribution transformers operate close to a load limit, which will be determined in this study. When baseloads are close to or above the load limit, ARVF administers low rates. Likewise, charging vehicles receive high rates when baseloads are not close to the load limit, reducing high peaks caused by uncontrolled charging and creating a valley-filling effect. The charging rate that is administered to vehicles is the optimization variable used in this study. The objective function defined by Equation (1),

$$\min F = \left| l_i - \left( \sum_{j=1}^{n} y_i + x_i \right) \right| \tag{1}$$

is used to determine the charging rate, $y_i$, delivered to PEVs at each 1 min timeslot expressed as $i$. This function ensures that the difference between the load limit curve denoted as $l_i$ and total load is minimized. The variables $l_i$ and $x_i$, the forecasted load, in the objective function are known variables that do not change during each timeslot. During each timeslot, ARVF analyzes the number of PEVs connected to the transformer and assigns them a charging rate based on their priority ratio, $p$, defined by Equation (2),

$$p = \frac{\text{Requested Charge (kWh)}}{\text{Remaining Dwell Time (minutes)}} \tag{2}$$

and the load difference ($\Delta$), defined by Equation (3) is the load limit minus the current baseload and is expressed in kW.

$$\Delta(i) = l_i - x_i \tag{3}$$

The amount of electricity PEVs receive is determined via a fuzzy logic approach. Fuzzy logic is a formal framework for simulating and executing human heuristic knowledge to regulate a system [51,52]. The four fundamental components of fuzzy logic are fuzzification, inference mechanism, rule-base, and defuzzification. Inputs are translated into fuzzy sets that are understood by the algorithm, the inference mechanism utilizes the rule-base to interpret inputs, and defuzzification reverts the output of the protocol back to a real number. Figure 2 depicts a fuzzy logic controller, as shown in Belohlavek and Klir [52] and Passino and Yurkovich [51], providing a more comprehensive definition of fuzzy logic.

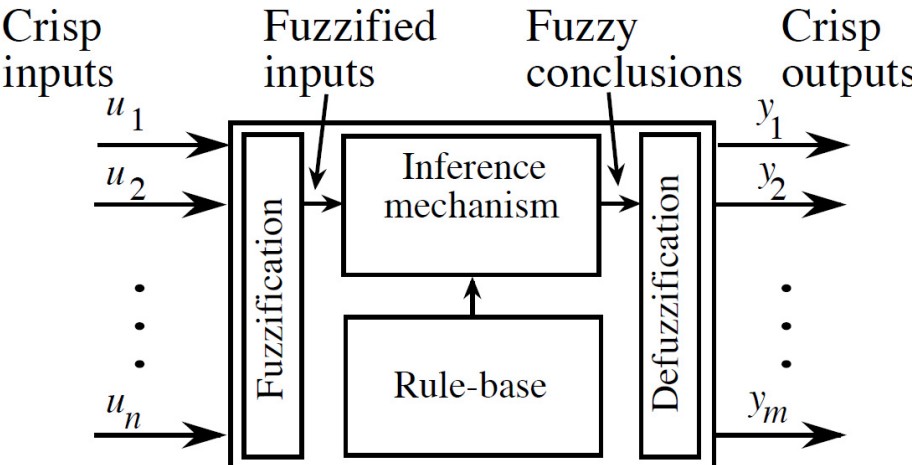

**Figure 2.** Fuzzy Logic Diagram.

The fuzzy logic decision mechanism used in this study begins by identifying the number of vehicles that are charging, the requested charge, and their remaining dwell time, to calculate $p$. The load difference, $\Delta$, is determined and used along with the priority ratio

as inputs to determine $y_i$ in Equation (1). The interpretation of load difference and each vehicle's priority ratio is determined by a set of criteria. The following framework is used to organize these rules,

$$\text{If premise, Then consequence}$$

This generalizes to

$$\text{If } u_1 \text{ is } A_1^j \text{ and } u_2 \text{ is } A_2^j, \text{ and } u_n \text{ is } A_n^j \text{ then } y_q \text{ is } B_q^p$$

where $y_q$ is an output variable interpreted as a linguistic variable and $B_q^p$ is a linguistic variable denoted by Equation (4).

$$B_i = \left\{ B_i^p : p = 1, \, 2, M_i \right\} \tag{4}$$

An expert specifies the sets of linguistic rules used to control the system to the desired state.

Both values are used as inputs and return an output of 1, 2, or 3, corresponding to small, medium, and large membership functions, shown in Figure 3. These classifications are used to group similar values of load difference and priority ratio together. The load difference membership functions are created using 1.9, 3.3, and 7.2 kW as the center of the three triangle membership functions, respectively. These three rates are used to determine if there is a small, medium, or large difference between the load and load limit. Ramos Muñoz et al. [46] use 3.3 and 7.2 kW in their study, and to maintain parallelism, both of these rates are used. An additional rate of 1.9 kW, is used because, in reality, charging rates below 3.3 kW exist. Not incorporating lower rates limits the algorithm's performance, reducing its ability to charge vehicles when the baseload is near the load limit. Through observation, values of 1, 3, and 5 were determined to be ideal numbers for the centers of the priority ratio membership functions.

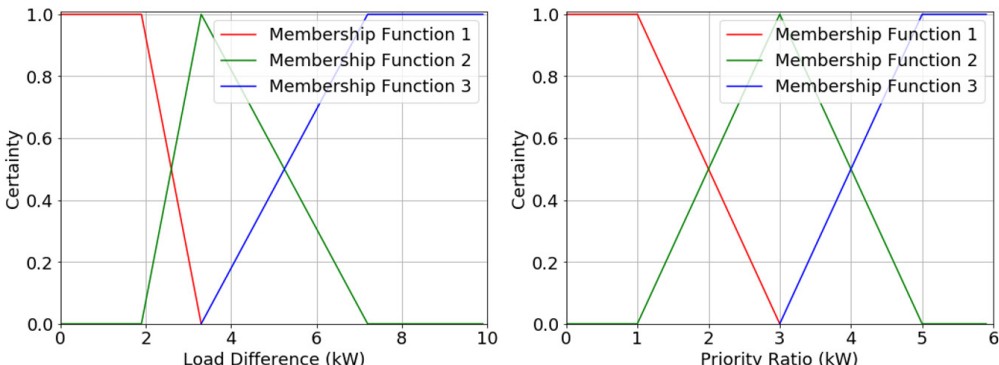

**Figure 3.** Load difference and priority ratio membership functions.

After the priority ratio and load difference are assigned to a membership function, an inference mechanism calculates the output membership function based on the load difference and priority ratio membership functions. The inference mechanism imitates the expert's decision-making process by analyzing which combination of if-then statements are satisfied and formulating a decision. The inference mechanism requires two steps. Step 1 is referred to as matching and utilizes the inference table, shown in Table 1, to assess which rules should be applied in each situation.

**Table 1.** Fuzzy logic interference table.

| Charge Rate | | Load Difference | | |
| --- | --- | --- | --- | --- |
| | | 1 | 2 | 3 |
| Priority Ratio | 1 | 1 | 2 | 3 |
| | 2 | 2 | 3 | 3 |
| | 3 | 2 | 3 | 3 |

The confidence of the outcome from step 1 is calculated in step 2. Step 2 is a two-step procedure that uses the minimum of input membership certainties, as indicated in Equation (5).

$$\mu_{premise_i} = \min\left(\mu_{A_i^j}(u_1),\ \mu_{A_i^j}(u_2),\ \mu_{A_i^j}(u_n)\right) \tag{5}$$

The implied fuzzy sets are derived by computing the membership function from the consequence (output membership function) and using the minimum to determine the "then" operation after calculating $\mu_{premise_i}$. The total implied fuzzy set, noted in Equation (6), is the result of combining these two implied fuzzy sets.

$$\mu_{(i)}(u) = \min\left(\mu_{premise_i},\ \mu_{B_i^j}(u_i)\right) \tag{6}$$

The numerical values 1, 2, and 3 are used to represent the output values of "small," "medium," and "high." For example, if the load differential is 3 (large) and a vehicle is assigned a priority ratio of 2 (medium), the vehicle will be charged at a rate of 3. (large). The output is assigned to a membership function, which calculates, $Y_i$, the output rate for the connected vehicle, using the "center of gravity" (COG) approach, as indicated in Equation (7).

$$Y_i = \frac{\sum_{k=1}^{R} b_k^i \int A}{\sum_{k=1}^{R} \int A} \tag{7}$$

where $R$ is the number of rules, $b_k$ is the center of the area of the output membership function, and

$$A = \int u_k(y_i)dy_i$$

is the area under the corresponding membership function. The output membership is depicted in Figure 4.

After the rate $Y_i$ is determined, the value is added to the transformer's current demand load for that time interval. This process is repeated for each charging PEV during this time interval, steadily increasing the demand. As the demand approaches the transformer's load limit, the rate that is delivered to each vehicle begins to decrease to protect the transformer. Table 1 illustrates this effect; low load difference values produce low delivered charging rates. After each charging PEV during the assessed time interval has been assigned a rate, and the demand increases to its final value, ARVF will evaluate the next time interval. A visual illustration of ARVF's process flow is depicted is Figure 5.

The load limit for ARVF is calculated by comparing seven cases, each with a distinct load limit. Each case will simulate charging PEVs on the baseloads described in data Section 2.3. The load limit that satisfies the primary objective (1) ensuring all vehicles charge greater than or equal to 90% of the total amount of energy that the vehicle requested, and minimizes the second objective (2) reducing the average transformer load during charging will be the finalized load limit for ARVF. These objectives are selected to satisfy the user demands and to optimize use of the available power [53].

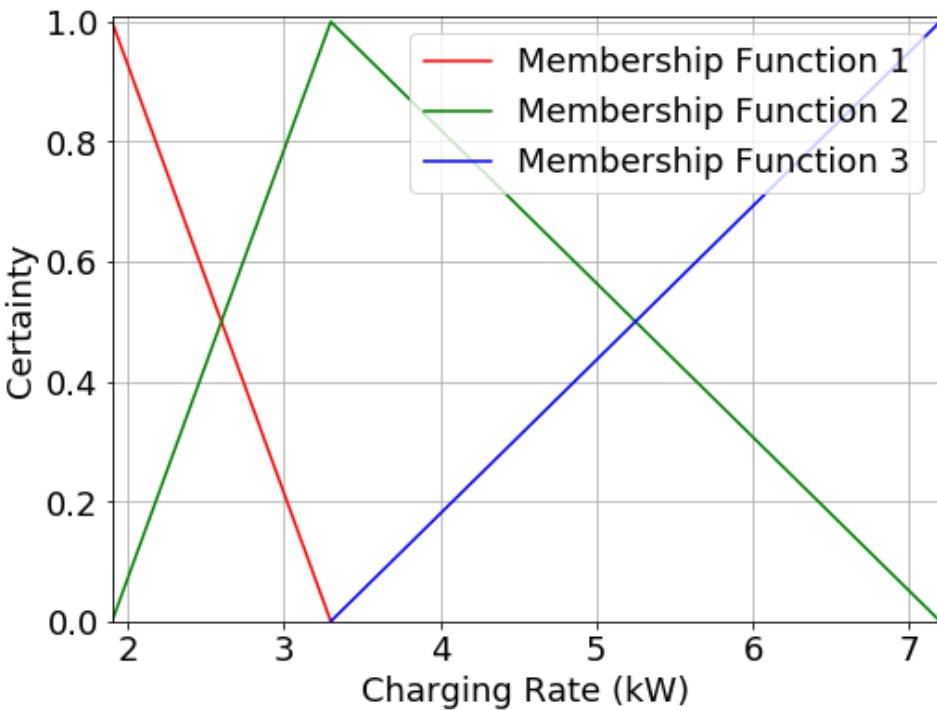

**Figure 4.** Output membership function.

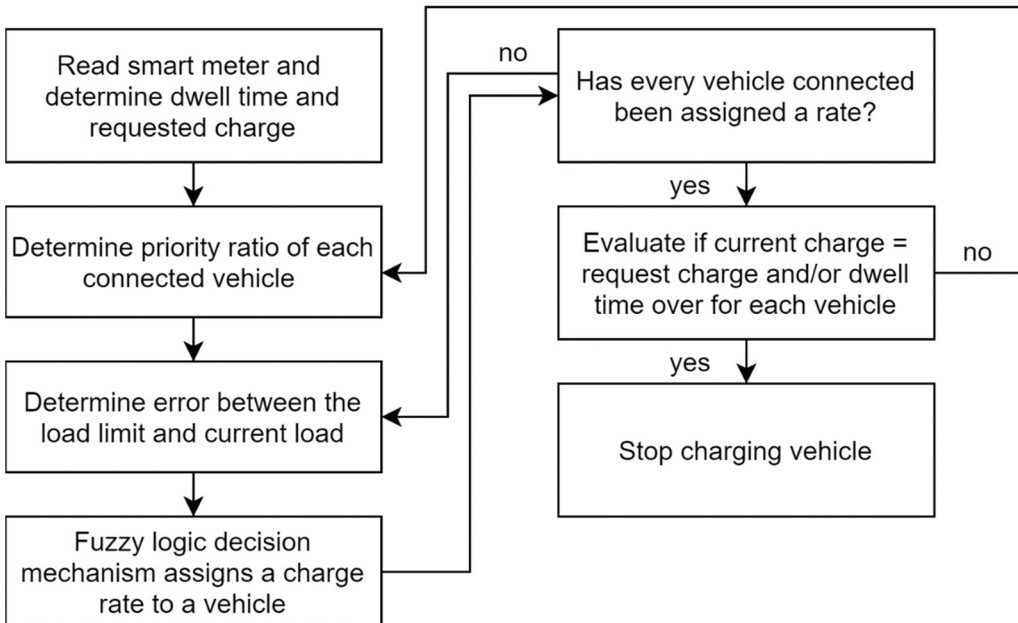

**Figure 5.** ARVF flow chart.

Razeghi et al. show that as the transformer load factor, a ratio of the observed load to the specified limit, rises, the winding hot spot temperature (HST) rises [19]. An aging acceleration factor (AAF) is used to quantify how much a varying load affects transformer life. AAF is calculated using Equation (8) according to the IEEE C57.91 standard, which shows that greater HST values increase the AAF [48].

$$\mathrm{AAF} = \exp\left(\frac{15,000}{383} - \frac{15,000}{\theta_{HST} + 273}\right) \tag{8}$$

For each time step, the AAF is calculated and used to determine the equivalent aging factor, EAF, shown using Equation (9).

$$\text{EAF} = \sum_{i=1}^{N} \text{AAF}_i \Delta t_i / \sum_{i=1}^{N} \Delta t_i \tag{9}$$

Loss of life percentage, shown in Equation (10), is then determined by multiplying EAF by the number of operational hours and dividing by normal insulation life, typically chosen to be 180,000 h.

$$\text{LOL\%} = \text{EAF} \times \sum_{i=1}^{N} \Delta t_i \times \frac{100}{180,000} \tag{10}$$

The goal of this project is to lower the transformer's loss of life percentage by focusing on HST reduction. It will be accomplished by lowering the average transformer load while charging, lowering the observed load and load factor, and lowering the HST as a result. The percentage that the average transformer load during charging is reduced by is used to assess the performance of each load limit and ARVF.

In addition, two more characteristics will be collected for data purposes but will not be utilized to determine the appropriate load limit: (1) the reduction in absolute maximum peak power achieved by all transformers and (2) the reduction in average maximum peak power reached by each transformer.

### 2.2. Transformer Data

The transformer data utilized in this study were captured on 25 September 2014, from a 75 kW home transformer in Irvine, California, and will be used to model the forecasted demand. The day's minimum and maximum temperatures were 22.2 degrees Celsius (72.0 degrees Fahrenheit) and 31.1 degrees Celsius (88.0 degrees Fahrenheit), respectively. The baseload on the transformer is the demand before any PEV charging demand is applied. The transformer referenced in this study provides power to 20 homes, whose square footage ranges from 1900 to 2900 square feet. In this analysis, electric vehicle charging was not included in the transformer baseload and data were sampled in 5 min intervals.

In this study, the demand curve that is used was recorded from midnight to midnight. The load profile is extended from 24 to 48 h, with the middle 24 h (hour 12 through hour 36) being used as the baseload, resulting in an overnight interval extending from midday to midday. This study uses the same transformer data as Ramos Munoz et al. [47], allowing for direct comparison and analysis. From hereon, this baseload will be referred to as the forecasted baseload.

The demand curves that are referred to as actual baseloads are generated by varying the forecasted baseload by a scaled percentage. A column vector of 48 by 1 is used to represent the 48 1 h sections in the baseload. A random number generator fills the cells with a 0 or 1, where cells that receive a 0 are decreased by a scaled percent value, and the cells that receive a 1 are increased by the same percentage. This is used to simulate the variation in an hourly fashion between the forecasted and actual baseload. Seven actual baseloads are analyzed, the variations ranging from 0% to 30% in 5% increments. Figure 6 below illustrates the forecasted (0% variation) and the six other actual baseloads used in this paper.

### 2.3. PEV Data

Data from the 2009 National Household Travel Survey (NHTS) are utilized to mimic automobile travel behavior, similar to the experiments described by Zhang et al. [19] and Ramos Munoz et al. [46]. The assumptions proposed by Ramos Munoz et al. [46] are implemented, resulting in 20,295 cars being randomly assigned to 2255 transformers, with a ratio of 9 PEVs per transformer. Throughout all simulations, each vehicle's initial randomized assignment is preserved.

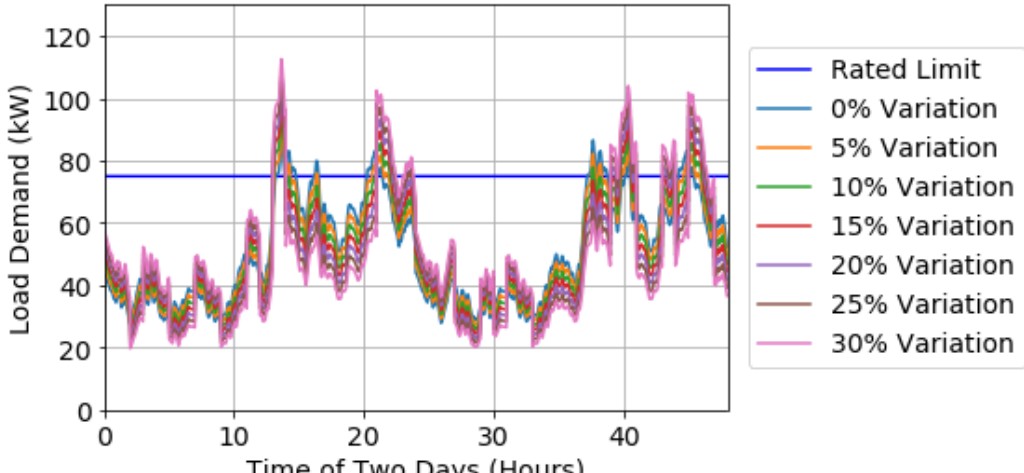

**Figure 6.** Demand curve with specified variation.

To simulate uncontrolled charging, the PEV data are superimposed on the transformer baseload. Uncontrolled charging is used as the baseline for this analysis to estimate the performance of controlled charging methods. When plugged in, all vehicles in the uncontrolled situation begin charging at a constant rate of 7.2 kW. The uncontrolled charging profiles are shown in Figure 7 before they are applied to the baseload. The demand from all 2255 transformers are represented by the green curves. The average load during charging, absolute maximum peak power reached amongst all transformers, and average maximum peak power reached by each transformer is 11.04, 50.40, and 23.31 kW, respectively.

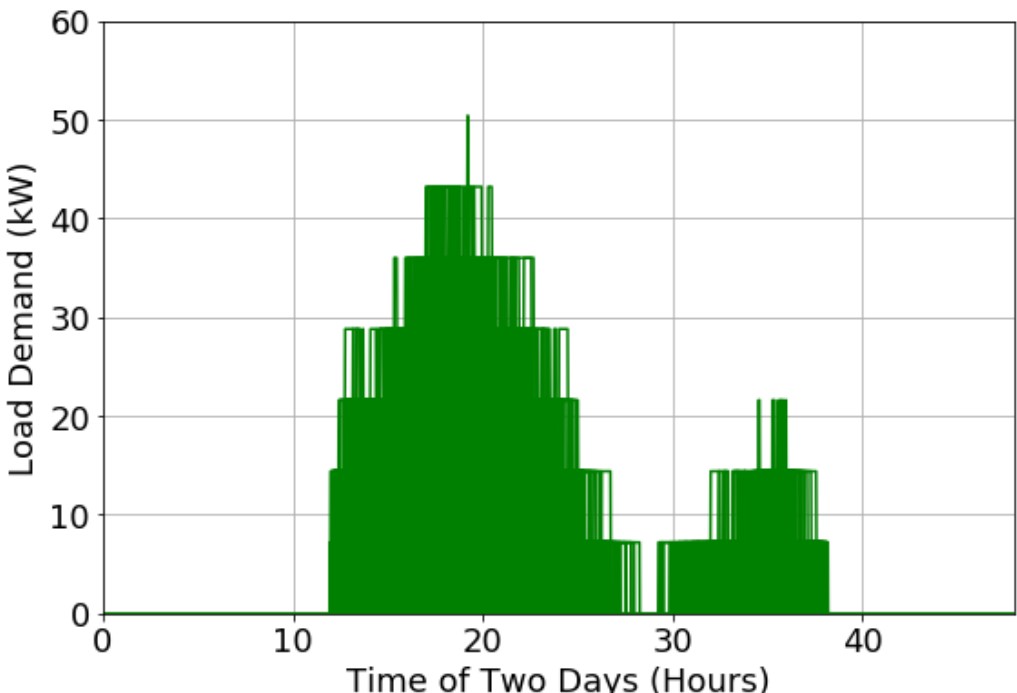

**Figure 7.** Uncontrolled charging profiles using 7.2 kW before they are applied to baseload.

## 3. Results and Discussion

ARVF utilizes a load limit to determine appropriate times to charge each vehicle and to compute the optimal rate each vehicle should receive. An analysis is conducted to explore 7 load limit cases and determine which is the best for ARVF. Stated previously in Section 2, the load limit that satisfies the primary objective (1) ensuring all vehicles charge greater than or equal to 90% of the total amount of energy that the vehicle requested, and minimizes the second objective (2) reducing the average transformer load during charging will be the finalized load limit for ARVF. After the load limit is validated, proving that it satisfies both objectives, ARVF will inherent this limit and operates by charging vehicles when the baseload is lower than the load limit.

In case 1, the load limit is equal to the rated limit, meaning that ARVF fills to 75 kW. In cases 2–7, the load limit is equal to the average baseload plus a multiple its standard deviation. In cases 2, 3, 4, 5, 6, and 7, the multiple of the standard deviation is equal to 0, 0.5, 0.75, 1, 1.5, and 2, respectively.

Cases 1, 2, and 3 do not charge all vehicles greater than or equal to 90% of the total amount of energy that the vehicle requested, shown in Table 2, and are eliminated from plausible load limits. Further analyzing cases 4, 5, 6, and 7, Table 3 shows each case's ability to reduce the average load during charging. Moreover, this table proves that case 4, equating the limit load to the average baseload plus 0.75 of its standard deviation performs best, reducing the average load during charging by 16.53%.

**Table 2.** Number vehicles charged in each case.

| Case | 0–40% | 40–50% | 50–60% | 60–70% | 70–80% | 80–90% | 90–100% |
|---|---|---|---|---|---|---|---|
| Uncontrolled | 0 | 0 | 0 | 0 | 0 | 0 | 20,295 |
| 1 | 0 | 0 | 0 | 0 | 0 | 0 | 20,295 |
| 2 | 76 | 51 | 56 | 64 | 51 | 57 | 19,940 |
| 3 | 0 | 0 | 0 | 0 | 0 | 1 | 20,294 |
| 4 | 0 | 0 | 0 | 0 | 0 | 0 | 20,295 |
| 5 | 0 | 0 | 0 | 0 | 0 | 0 | 20,295 |
| 6 | 0 | 0 | 0 | 0 | 0 | 0 | 20,295 |
| 7 | 0 | 0 | 0 | 0 | 0 | 0 | 20,295 |

**Table 3.** Demand changes in each case.

| Case | Average Load during Charging | Average Load during Charging Percent Difference |
|---|---|---|
| Uncontrolled | 74.36 | - |
| 1 | 47.54 | 36.07% |
| 2 | 50.62 | 31.93% |
| 3 | 58.35 | 21.53% |
| 4 | 63.16 | 16.53% |
| 5 | 65.75 | 12.29% |
| 6 | 71.09 | 4.50% |
| 7 | 74.03 | 0.44% |

Case 4 is selected as the limit load for ARVF because it performs optimally amongst the 7 cases, successfully charging all vehicles greater than or equal to 90% of the total amount of energy that the vehicle requested and reducing the average load while charging the most. This investigation shows that the rated limit is not an adequate load limit, case 1, as not all cars are able to charge greater than or equal to 90% when ARVF is applied

to the baseload load limit. Superior results are obtained by equating the load limit to a value that is relative to the baseload. A load limit that can adjust to the baseload allows the algorithm to scale accordingly, preventing (1) small-scaled uncontrolled charging from occuring when the baseload is significantly below the rated limit and (2) low amounts of vehicles reaching full charge when the baseload is significantly above the rated limit. Based on the results in Table 3, the load limit in case 4 will be used to assess this algorithm because case 4 performs the best amongst the 7 presented cases.

Ramos Muñoz et al. [47] evaluated six algorithms, including Grid Valley filling by Zhang et al. [19] and Time-Of-Use (TOU) Charging from Southern California Edison (SCE) [54]. This study determined that the Grid Valley Filling with Modified Timeslot Rejection strategy produces the best results, preventing all local transformers from experiencing significant overloading. Grid Valley Filling with Modified Timeslot Rejection will hereby be referred to as MTR and will be used to evaluate the effectiveness of ARVF.

When ARVF and MTR are applied to the forecasted baseload (0% variation), shown in Figure 8, the absolute maximum peak power reached amongst all transformers, the average maximum peak power reached by each transformer, and maximum peak power from the baseload are all 86.58 kW. This signifies that the maximum peaks are produced by the baseload rather than PEV charging from the algorithms. The average load during charging when ARVF and MTR are applied is 63.13 and 54.81 kW, respectively. MTR performs very well because it is completely reliant on the forecasted load, and in this situation, the actual load is the forecasted load.

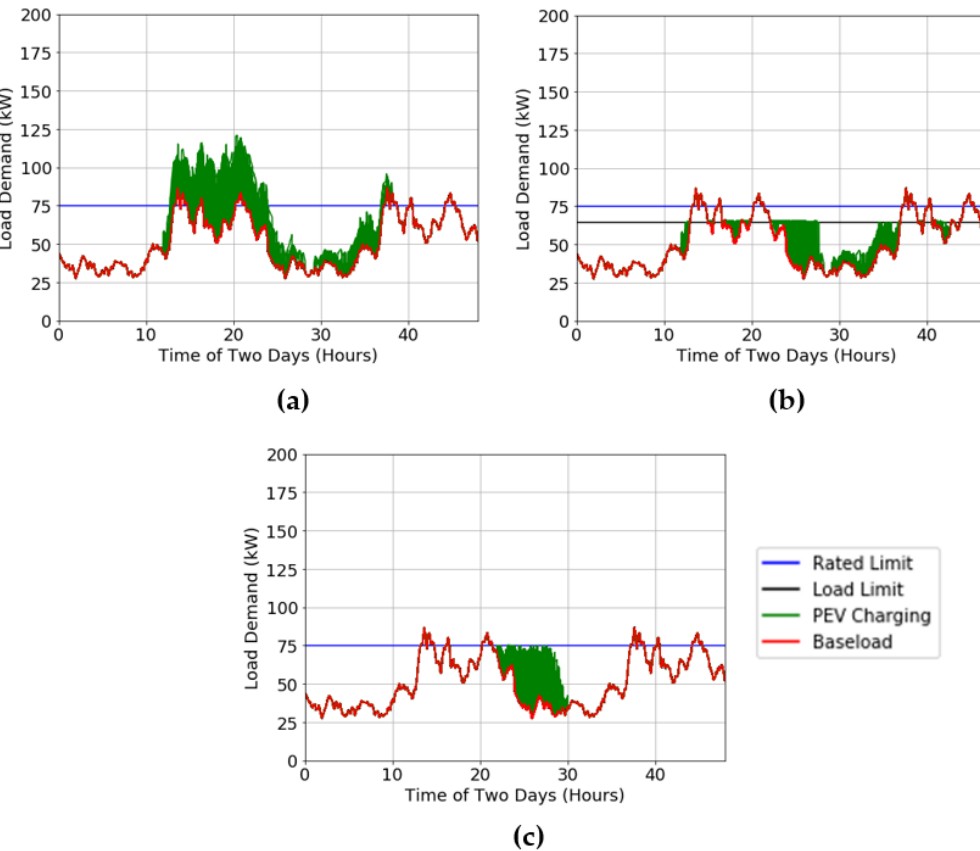

**Figure 8.** Charging profiles created from (**a**) uncontrolled charging, (**b**) ARVF and (**c**) MTR on the 0 percent variation baseload.

In each subplot in Figure 8, there are 2250 green curves; each curve represents the total load observed by one transformer. In Subplot A, vehicles are charging without any scheduling protocol (uncontrolled charging), meaning as soon as each vehicle arrives home, it immediately begins charging. This creates a lot of variation between the 2250 green

curves, generating little overlap between the curves and appearing as if a lot of charging is occurring because there is a lot of visible green shading. On the contrary, in subplots B and C, vehicles are charging using a valley-filling algorithm. Most of the vehicles begin charging in Subplots B and C are generally shifted to similar times, despite when each vehicle arrives at their perspective home. This creates very little variation between the 2250 green curves, generating a significantly large overlap between the curves and appearing as if very little charging is occurring because there is a small amount of visible green shading. If each curve from every transformer in Subplot A is compared to the curve produced by the same transformer in Subplot B or C, the area under both curves would be the same. The shape of the curves in Subplots B and C are generally much flatter and wider than in Subplot A but represent the same amount of charging.

As the variation begins to increase and the actual and forecasted demand deviate (i.e., a day with more homes using an air conditioner than expected), ARVF's performance increases, and MTR's performance decreases. ARVF determines how much charge a vehicle receives in a real-time, per-minute fashion based on the load the transformer is observing in that time interval. This attribute allows ARVF to adapt to unforeseen changes; if the baseload suddenly increases, the algorithm will respond to the change and adjust the amount of charge being delivered to vehicles. In this same instance, the Modified Timeslot Rejection strategy is unable to adapt to changes in the forecasted baseload and therefore uses the same charging profile for a now different baseload. When the actual baseload has approximately 20% variation from the forecasted baseload, the algorithms generally perform the same. The charging distributions are displayed in Figure 9.

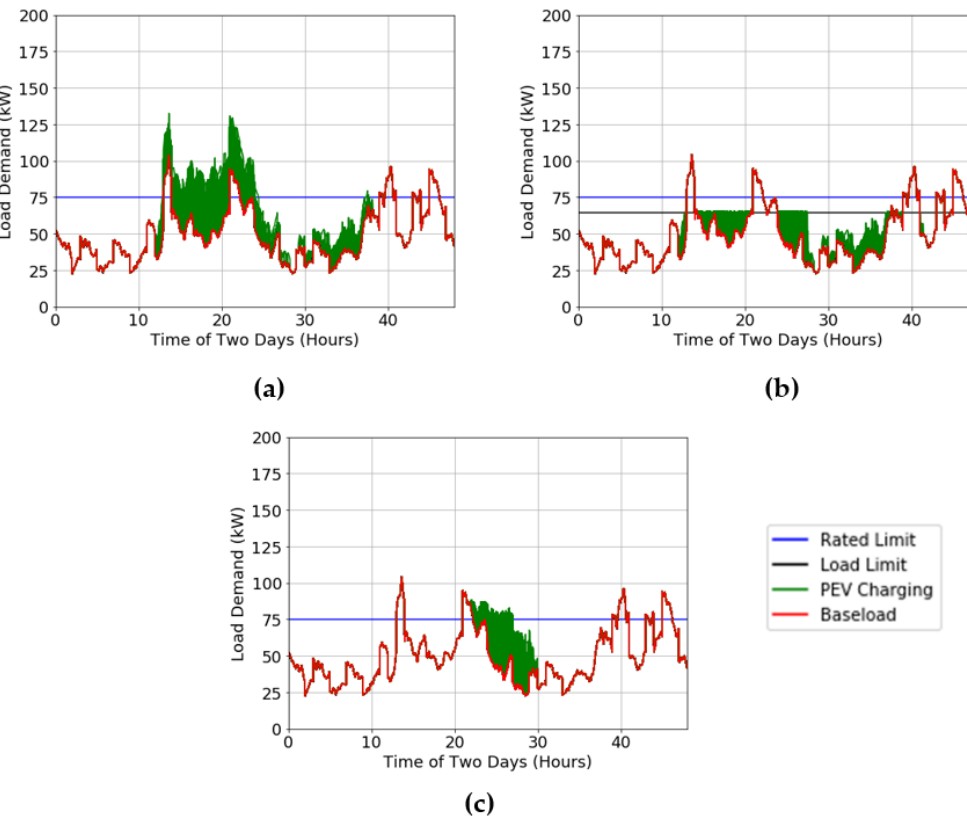

**Figure 9.** Charging profiles created from (**a**) uncontrolled charging, (**b**) ARVF and (**c**) MTR on the 20 percent variation baseload.

Figure 10 depicts when 30% variation is present between the actual and forecasted curves. Despite the large change in the two curves, MTR does not respond to this change, maintaining the original charging sequence and scheduling vehicles in the same timeslots from the decision on the outdated forecasted curve. In contrast, ARVF reacts to the change by delivering lower rates to the charging vehicles and extending the duration of the charge. In the 0% variation example, the actual and forecasted baseload match perfectly, making the time MTR scheduled to charge vehicles ideal and outperforming ARVF. However, Table 4 shows that when the actual and forecasted baseload differs by 20% or more, ARVF's produces better results due to its real-time robustness.

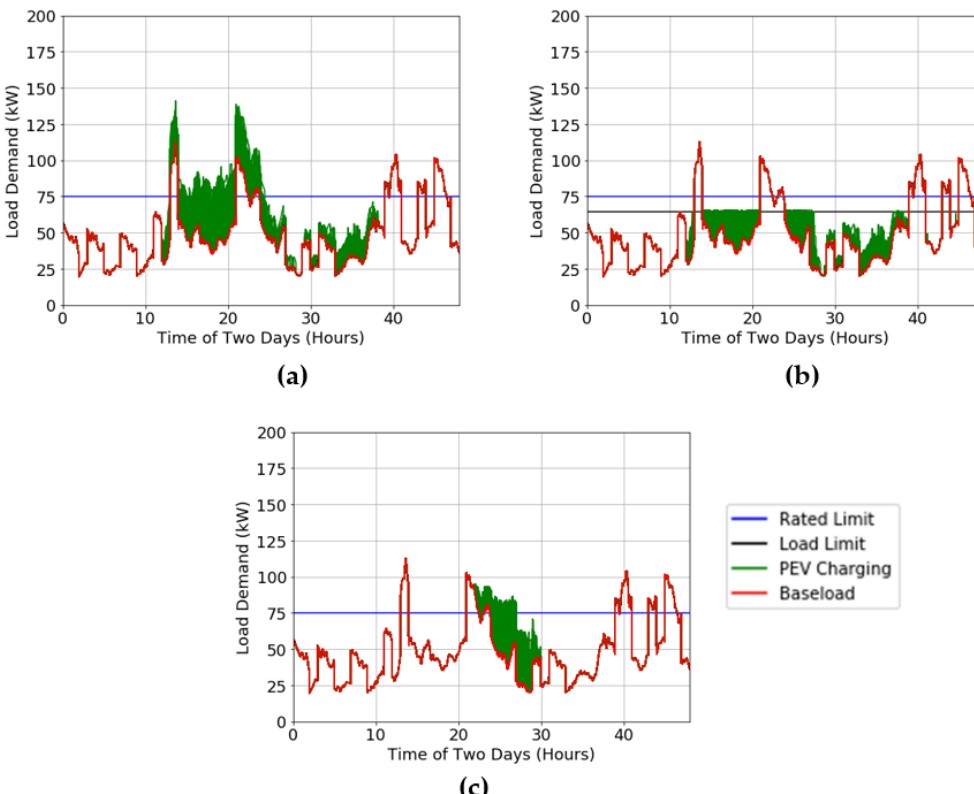

**Figure 10.** Charging profiles created from (**a**) uncontrolled charging (**b**) ARVF and (**c**) MTR on the 30 percent variation baseload.

Table 4 shows that as variation is introduced to the forecasted and actual baseload, MTR's advantage steadily diminishes. In addition, Table 4 shows that ARVF can achieve similar results to MTR and allows for variation between the forecasted and actual baseload, making this algorithm well suited for real-world applications. In real-world applications, the forecasted load will rarely match the baseload, thereby highlighting the efficacy of this method.

**Table 4.** Algorithm comparison.

| Profile | Variation Percentage | Absolute Maximum Peak | Average Maximum Peak | Average Load during Charging | Absolute Maximum Peak Percent Difference | Average Maximum Peak Percent Difference | Average Load during Charging Percent Difference |
|---|---|---|---|---|---|---|---|
| Uncontrolled | | 120.82 | 97.87 | 74.53 | - | - | - |
| ARVF | 0 | 86.58 | 86.58 | 63.16 | 33.01832 | 12.2418 | 16.51536 |
| MTR | | 86.58 | 86.58 | 54.81 | 33.01832 | 12.2418 | 30.49327 |
| Uncontrolled | | 119.57 | 97.85 | 73.18 | - | - | - |
| ARVF | 5 | 90.91 | 90.91 | 62.89 | 27.23299 | 7.353253 | 15.12457 |
| MTR | | 90.91 | 90.91 | 55.86 | 27.23299 | 7.353253 | 26.84439 |
| Uncontrolled | | 123.89 | 101.03 | 71.81 | - | - | - |
| ARVF | 10 | 95.24 | 95.24 | 61.98 | 26.14886 | 5.900036 | 14.69467 |
| MTR | | 95.24 | 95.24 | 56.91 | 26.14886 | 5.900036 | 23.15103 |
| Uncontrolled | | 128.22 | 105.10 | 70.47 | - | - | - |
| ARVF | 15 | 99.57 | 99.57 | 60.89 | 25.15475 | 5.403821 | 14.58587 |
| MTR | | 99.57 | 99.57 | 57.97 | 25.15475 | 5.403821 | 19.46434 |
| Uncontrolled | | 132.54 | 109.28 | 69.12 | - | - | - |
| ARVF | 20 | 103.90 | 103.90 | 59.84 | 24.22602 | 5.047378 | 14.39206 |
| MTR | | 103.90 | 103.90 | 59.02 | 24.22602 | 5.047378 | 15.76401 |
| Uncontrolled | | 136.86 | 113.47 | 67.77 | - | - | - |
| ARVF | 25 | 108.23 | 108.23 | 58.45 | 23.36285 | 4.727109 | 14.76787 |
| MTR | | 108.23 | 108.23 | 60.07 | 23.36285 | 4.727109 | 12.04631 |
| Uncontrolled | | 141.18 | 117.67 | 66.42 | - | - | - |
| ARVF | 30 | 112.56 | 112.56 | 56.68 | 22.55852 | 4.439039 | 15.82453 |
| MTR | | 112.56 | 112.56 | 61.12 | 22.55852 | 4.439039 | 8.311118 |

## 4. Conclusions

ARVF was created as a strategic way to decrease the enormous demand on household transformers caused by unregulated PEV charging. This is desirable since decreased demand reduces stress and increases transformer lifetime [15,19]. By monitoring the baseload and load limit, ARVF assigns rates to cars using fuzzy logic. According to the findings of this study, ARVF functions best when the load limit is equal to the average value plus 0.75 of its standard deviation.

ARVF was designed to adjust to unforeseen variations in the forecasted baseload. It is compared to Ramos Muñoz et al. [46] Grid Valley Filling with Modified Timeslot Rejection algorithm as it has already been shown to be an improvement over several others [42]. Results show that in situations when the actual and forecasted baseload diverge, ARVF is robust and adjusts well to unforeseen variations. Moreover, this study proves that when the forecasted and actual baseload vary by more than 20%, ARVF can produce better results than the Modified Timeslot Rejection algorithm.

To improve the algorithm's performance, researchers should look at how much of the baseload should be approximated before execution and how this affects ARVF's capacity to valley fill. The load limit employed in this study was calculated using one baseload with multiple PEV charging profiles. Further baseload and demand combinations should be evaluated to validate this number. A continual analysis should be undertaken to evaluate if the load limit determined in this study should be modified to account for baseloads

changing due to transformers evolving, additional entities connecting to the grid, and devices requiring more power.

Because of its intricacy and real-time use, this algorithm is more suited to be implemented in the future. To successfully install a controller on residential transformers, the power distribution hardware must be upgraded. A computerized network infrastructure that connects smart chargers to a central distribution transformer and allows for bi-directional communication needs to be implemented. In addition, a fuzzy logic algorithm may be difficult to implement due to the complexity of the electrical system; however, replacing fuzzy logic with crisp logic reduces the complexity and accelerates the transformation of this idea from concept to reality. Crisp logic simplifies the process by allowing ARVF to deliver preset discrete charging rates to PEVs, whereas fuzzy logic allows ARVF to deliver any charging rate within a defined spectrum to PEVs. This concept illustrates how technology can be implemented in the future to allow algorithms to control residential transformer demand loads and, more importantly, a depiction of how future transformers and PEV chargers can interact together. This work aims to demonstrate the benefit of what can be achieved if the infrastructure to support this idea is created.

**Author Contributions:** Conceptualization, T.S., J.G. and G.W.; methodology, T.S. and J.G.; software, T.S.; validation, T.S. and J.G.; formal analysis, T.S.; investigation, T.S.; resources, T.S.; data curation, T.S.; writing—original draft preparation, T.S., J.G. and G.W.; writing—review and editing, T.S., J.G. and G.W.; visualization, T.S.; supervision, G.W.; project administration, G.W. All authors have read and agreed to the published version of the manuscript.

**Funding:** This research received no external funding.

**Institutional Review Board Statement:** Not applicable.

**Informed Consent Statement:** Not applicable.

**Data Availability Statement:** Code generated during this study to simulate and assess the proposed algorithm is proprietary and confidential in nature and may only be provided with restriction.

**Conflicts of Interest:** The authors declare no conflict of interest.

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
