# Peer review of "Smart Electric Vehicle Charging via Adjustable Real-Time Charging Rates"

_applsci, doi:10.3390/app112210962_

Round 1

Reviewer 1 Report

In this paper, authors present a PEV charging algorithm, CVVF, to minimize the adverse effects from charging on the grid. And this algorithm is more robust. There are some issues in this paper

  1. Many typos. For example, p2.65 vehicle to gas(grid); p4 equation(1), ‘j’ is a typo? p8.267 Each cause(case). Authors should be more careful.

  1. There are many variables in this algorithm, but the expression in section 2 is a little confusing, it is hard to know the order of calculation. This should be improved, for example, by using a more detailed flowchart to describe the relationship between these variables.

  1. In abstract, authors said ‘more advantageous than other optimization techniques requiring significantly more computation’. but in the paper, the CVVF is only compared with MTR. MTR looks like a more concise algorithm. ‘significantly more computation’ is an unconvincing description. Authors should show the calculate time of these algorithms.

Reviewer 2 Report

The paper, which presents a EV control methodology, is well written and interesting to read. The paper addresses one of the most significant issues associated with EV grid integration, which is the impact on the secondary system and transformers.

I noticed minor typos at: (1) line 28 i wasn't sure why PEVs was used twice; (2) line 300 the second Fahrenheit should be Celsius; (3) i think "is used" should be removed in line 307

It wasn't clear to me where "Membership Functions" came from and how it is used. Is it possible to provide more background? One element of confusion, for example, was Fig. 3 has load difference numbers between 0 and 10, but the text at line 230 only discusses values 1, 2, and 3.

I was also confused by the 7 different load limit cases. Where did those come from? 

I thought Figures 8, 9, and 10 were very interesting. Do these plots describe the PEV charging by filling the green color inbetween the baseload and the limits of the PEV charging? Or is there significant variability where the load is quickly going up and down? 

I gather that this approach works better than others because it doesn't depend on an accurate forecast as much as other methods. Correct? Can you comment on why that is important? What is the state of the art of EV forecasting at the transformer level?

Reviewer 3 Report

The paper deals with very actual topic of reducing the impact of EV chargers to the distribution grid. The results presented in the paper show the applicability of the CVVF methodology. However some issues require more detailed explanation to properly understand the method.

  1. The introductory section gives a literature review of existing methodologies, but the contribution of the paper is focused to the comparison with MTR methodology only. Some advantages of CVVF method are emphasized at the middle of the introductory section, other at the end.

The suggestion is to present the existing methodologies first. Secondly, based on the analysis of all presented research, to point out the difference between them, the advantages and the overall contribution of the paper.

Finally, the structure of the paper should be given at the end of the introductory section.

  1. Some mistakes in the introductory section should be corrected: V2G means „vehicle to grid“ no „vehicle to gas“.
  2. „Prehistoric data“ should be changed to „historic data“.
  3. The section 2 gives some general and very known facts about the fuzzy reasoning, and this part can be omitted.
  4. Authors should explain how the membership functions on Figure 3 were constructed and how the presented triangular shapes are selected.
  5. Section 3 is missing, or the Section 4 should be renumbered.
  6. Generally, it is not clear why the fuzzy logic is introduced at all instead of other control algorithms.
  7. Why the forecasted data are important if the operator continuously vary the charging rate and the real transformer values are known. The forecasted value could have an impact on projected transformer heating, but this aspect is not present in the paper.
  8. The conclusion section is confusing. Authors say that „Crisp logic simplifies the process by allowing CVVF to deliver variable charging rates to PEVs, whereas fuzzy logic allows CVVF to deliver variable charging rates to PEVs“. What is the difference?
  9. Some grammatical errors should be corrected as well („number vehicles“, „CVVF can be charge vehicles“... etc)

Reviewer 4 Report

This is a nice study on EVs. The authors should consider the following.

  1. BEVs FCEVs and PEVs are not zero-emission vehicles. Electricity and hydrogen are not primary energy sources and are produced from other fuels. For example, if electricity is generated from coal and hydrogen from petroleum there is a lot of CO2 emissions to be attributed to EVs. A good explanation is in:  Michelides, E. E., 2020, Thermodynamics and Energy Consumption of Electric Vehicles, Energy Conversion and Management,vol. 203, 112246. 
  2. Check your data of Figure 1, they are not correct. Get accurate data from the US-DOE website.
  3. A large part of the introduction refers to the power drawn (kW) and not to the energy (kWh). The authors should mention at least how many hours the EVs are charging in their time management sections.
  4. Explain in detail "...to dramatically degrade transformer lifespan." How about the grid capacity?
  5. Can fuzzy logic and the algorithm respond to emergencies? What if the EV NEEDS to be used at a certain time.
  6. The last paragraph in the conclusions essentially tells that this system is impossible to implement. Is this the authors' intention? If not, can they provide information on how this system may become a reality? 

Round 2

Reviewer 3 Report

Authors answered to the majority of questions and improved the readability of the paper. However, some issues are still requiring the more detailed explanation.

  1. Concerning the question related to the way the triangular fuzzy number is formed, it would be more appropriate to represent the load difference and priority ratio in relative units. In Figure 3, load difference is probably expressed in kW (although the units are not represented!) and linked to the 75 kW transformer load, greatly reducing the generality of the proposed method.
  2. As far as priority ratio is concerned, the unit is not given neither in expression (2) nor in Figure 3.
  3. The reasons to use the different forecasted loads are still unclear. In AVFR flowchart (figure 5) the real time charging rate is determined using the smart meter data. Where the forecasted value fits in this algorithm?
  4. What the objective to charge the vehicle above 90% means? Is it the state of charge?
  5. How the objective to charge every vehicle above 90% fits in the algorithm in figure 5?
  6. The energy for charging the vehicles (green surface in Figure 8 a) is greater than the charging energy in figures 8 b) and 8 c). How the authors explain this fact, claiming at the same time that every vehicle is charged above 90%?

Reviewer 4 Report

The authors have not addressed most of my comments. They just supply links to questionable websites.

Do they really think that AVs have zero emissions?

The charging time of EVs and the energy charge are directly related to the needed power.

Still the "fuzzy logic" approach is incompatible with an electrcicity grid (reliability issues).

And several others...

Round 3

Reviewer 3 Report

Authors answered to all of my questions. I have no furhter suggestions.

Author Response

No need to do anything further. The reviewer said, " The authors answered all of my questions. I have no further suggestions."